# Enhancement of Immune Checkpoint Inhibitor-Mediated Anti-Cancer Immunity by Intranasal Treatment of *Ecklonia cava* Fucoidan against Metastatic Lung Cancer

**DOI:** 10.3390/ijms22179125

**Published:** 2021-08-24

**Authors:** Wei Zhang, Juyoung Hwang, Dhananjay Yadav, Eun-Koung An, Minseok Kwak, Peter Chang-Whan Lee, Jun-O Jin

**Affiliations:** 1Shanghai Public Health Clinical Center, Shanghai Medical College, Fudan University, Shanghai 201508, China; weiwei061215@126.com (W.Z.); jyhwang5@yu.ac.kr (J.H.); 2Department of Medical Biotechnology, Yeungnam University, Gyeongsan 38541, Korea; dhanyadav16481@gmail.com (D.Y.); eunkoungan@yu.ac.kr (E.-K.A.); 3Research Institute of Cell Culture, Yeungnam University, Gyeongsan 38541, Korea; 4Department of Chemistry, Pukyong National University, Busan 48513, Korea; mkwak@pukyong.ac.kr; 5ASAN Medical Center, Department of Biomedical Sciences, University of Ulsan College of Medicine, Seoul 05505, Korea

**Keywords:** *Ecklonia cava* fucoidan, mucosal adjuvant, anti-PD-L1 antibody, immunotherapy, anti-cancer

## Abstract

Although fucoidan, a well-studied seaweed-extracted polysaccharide, has shown immune stimulatory effects that elicit anticancer immunity, mucosal adjuvant effects via intranasal administration have not been studied. In this study, the effect of *Ecklonia cava*-extracted fucoidan (ECF) on the induction of anti-cancer immunity in the lung was examined by intranasal administration. In C57BL/6 and BALB/c mice, intranasal administration of ECF promoted the activation of dendritic cells (DCs), natural killer (NK) cells, and T cells in the mediastinal lymph node (mLN). The ECF-induced NK and T cell activation was mediated by DCs. In addition, intranasal injection with ECF enhanced the anti-PD-L1 antibody-mediated anti-cancer activities against B16 melanoma and CT-26 carcinoma tumor growth in the lungs, which were required cytotoxic T lymphocytes and NK cells. Thus, these data demonstrated that ECF functioned as a mucosal adjuvant that enhanced the immunotherapeutic effect of immune checkpoint inhibitors against metastatic lung cancer.

## 1. Introduction

Marine seaweed polysaccharides exhibit immunomodulatory effects. Recently, sulfated polysaccharides, such as fucoidan, carrageenan, ascophyllan, porphyran, and ulvan have been explored as vaccine adjuvants for the treatment of cancer and infectious diseases and are widely used in the pharmaceutical industry [1,2,3,4,5,6,7,8]. Several biological actions of fucoidans have been reported, including antioxidant, anti-inflammatory, anti-hepatic, anti-atherosclerotic, and anti-allergic activities [9,10]. Moreover, the anti-tumor effects of fucoidans have been well illustrated because of their interaction with various signaling mechanisms [11,12,13,14]. Fucoidans extracted from various sources, such as *Fucus vesiculus*, *Fucus evanescens*, *Laminaria japonica*, and *Macrocystis pyrifera* have stimulatory effects on dendritic cells (DCs), macrophages, T cell lymphocytes, and other immune cells [15,16,17,18]. A previous study showed that fucoidan from *Ecklonia cava* (ECF) also induces activation of immune cells [19]. Our group has also shown that ECF contained a higher amount of fucose, galactose and sulfate ester group than *Macrocystis pyrifera* fucodian, *Undaria pinnatifida* fucoidan and *Fucus vesiculosus* fucoidan [18] and it promotes the strongest effect of immune cell activation compared to those fucoidans in human peripheral blood mononuclear cells [20]. In addition, intraperitoneal administration with ECF activates splenic DCs in the mouse in vivo, which elicits antigen-specific immune responses by the ECF and ovalbumin in the mice [7]. Therefore, ECF could be used as an adjuvant to promote anticancer properties. However, the mucosal adjuvant effect of ECF, especially activation of respiratory track immunity, has not been investigated.

Metastatic cancer cells from a primary tumor to distant sites are the main reason for human cancer-related deaths [21]. Some tumors exhibit very strong migration ability, which forms metastatic cancer lesions with high frequency including melanoma, colorectal carcinoma, and small cell carcinoma of the lung [22]. The lung can be a target organ for metastasis of carcinoma and melanoma [23,24]. Despite a variety of approaches for lung metastasis treatment, such as chemotherapy, radiotherapy, and targeted therapy, the survival rate of metastatic lung cancer remains very low [24]. Since selected patients have a chance of surgery for metastatic lung cancer, immunotherapies have been gaining attention for treatment of metastatic lung cancer [24].

Cytotoxic T lymphocytes (CTLs) and natural killer (NK) cells are well known as the main contributors to cancer immunotherapy [24,25,26,27]. The stimulation of these CTLs and NK cells is mediated by antigen-presenting cells, especially DCs [20,28]. Mucosal surfaces covered by mucus are physical barriers that inhibit vaccines and adjuvants from reaching mucosal epithelial cells, thereby blocking immune responses of antigen-presenting cells (APCs) [29,30]. It is crucial to activate APCs in mucosal tissues through the co-administration of vaccines and adjuvants for effective treatment in preventing infectious diseases and mucosal cancers [31]. However, there are limited numbers of adjuvants that promote effective activation of immune cells for mucosal cancer treatment.

Previous studies of fucoidan in adjuvant settings in combination with anti-cancer drugs have demonstrated that fucoidan may be able to reduce the anti-cancer drug-mediated toxicity [32,33]. Programmed cell death-1 (PD-1) signaling plays a crucial role in therapeutic purposes in cancer immunotherapy and is a well-known target in the development of anti-cancer drugs [34,35]. PD-1 signaling pathways promote cancer development and progression by elevating the survival capacity of cancer cells [34]. Blockage of PD-1 signaling considerably improves antitumor immunity [36,37]. However, the inhibition of PD-1 and PD-L1 binding showed limited effect against metastatic lung cancer [38,39,40]. Therefore, in this study, we hypothesized that the intranasal treatment with ECF may enhance the anti-cancer activity of anti-PD-ligand 1 (anti-PD-L1) antibody for elimination of metastatic lung cancer. We carried out the present study to examine the hypothesis that the enhancement of immune checkpoint inhibitor-mediated anti-cancer effect by intranasal administration of ECF against melanoma and carcinoma in mice.

## 2. Results

### 2.1. Chemical Analysis of ECF

Since the molecular weight, homogeneity, and monosaccharide composition of ECF are some of important information on biological activity, the content of extracted ECF was analyzed by gel permeation chromatography (GPC) and high-performance liquid chromatography (HPLC). The weight-average molecular weight of ECF was approximately 42.95 kDa and polydispersity index (PDI) was 1.39 as homogeneity (Table 1). The monosaccharide composition of ECF determined that the major sugar was fucose (45.13%) and galactose (23.24%), whereas xylose (1.78%), glucose (1.37%), and mannose (0.55%) contents constituted a minor component, which was consistent with previous studies [20,41]. Furthermore, Fourier-transform infrared spectroscopy (FT-IR) spectral peaks of ECF revealed that the presence of sulfate groups at 823.51 cm^−1^ for C-O-S and at 1217.82 cm^−1^ for S=O (Appendix A).

### 2.2. ECF Induced Stimulation of DC in the mLN

To evaluate the immune stimulatory effect of ECF in the lung, C57BL/6 mice received 50 mg/kg ECF intranasally (*i*.*n*.) (Figure 1A). The DCs in mediastinal lymph node (mLN) were defined as shown in Figure 1B and analyzed the activation of the DCs by ECF. Intranasal treatment with ECF significantly increased the population and frequency of mLN DCs 18 h after treatment (Figure 1C), which was consistent with the increased number of DCs in the mLN compared to that of the phosphate buffered saline (PBS) treatment (Figure 1D). Moreover, expression levels of C-C chemokine receptor type 7 (CCR7) were substantially upregulated by ECF treatment in mLN DCs compared to that of the control, which indicated that ECF promoted the migration of DCs to the mLN (Figure 1E). In addition, ECF elicited upregulation of co-stimulators, major histocompatibility complex (MHC) class I, and II expression in mLN DCs (Figure 1F). The dose dependent administration of ECF showed that 50 mg/kg of ECF was optimal concentration for induction of DC activation in the mLN (Appendix A). The producing levels of interleukin (IL)-6, IL-12p70, and tumor necrosis factor (TNF)-α in bronchoalveolar lavage (BAL) fluid were also substantially elevated by ECF compared to that in the PBS-treated control (Figure 1G).

Since the *i*.*n*. administration of ECF promoted DC activation in mLN, we also examined whether the ECF can promote the DC activation in BALB/c mice. As shown in Appendix A, the *i*.*n*. treatment with ECF elicited great increases in the levels of co-stimulators and MHC molecules in DCs as well as pro-inflammatory cytokines in BAL fluid in the BALB/c mice.

### 2.3. ECF Induced Activation of NK Cells in the mLN

Because mature DCs promote different immune cell activation [27,42], whether *i*.*n*. treatment with ECF induces NK cell activation in the mLN was examined. NK cells in the mLN were defined as NK1.1^+^CD3^−^ cells in C57BL/6 mice (Figure 2A) and CD49b^+^TCR-β^−^ cells in BALB/c mice (Appendix A). The frequency and population of NK cells in mLN of C57BL/6 were dramatically increased with *i*.*n*. administration of ECF (Figure 2B). Consistent with the increased frequency and population of NK1.1^+^CD3^−^ cells, the number of NK cells in the mLN of C57BL/6 mice was also dramatically elevated by ECF treatment compared to PBS treatment (Figure 2C). In BALB/c mLN, it also showed great increases in the number of NK cells by *i*.*n*. treatment with ECF compared to PBS-treated control (Appendix A). In both C57BL/6 and BALB/c mice, *i*.*n*. administration of ECF promotes upregulation of CD69 and tumor necrosis factor-related apoptosis-inducing ligand (TRAIL) in mLN NK cells, which are the activation surface markers of NK cells (Figure 2D and Appendix A). The levels of interferon (IFN)-γ, granzyme B, and perforin produced in the BAL fluid of C57BL/6 mice were also greatly increased by ECF treatment (Figure 2E). The efficacy of NK cell activation by ECF in the mLN was similar to lipopolysaccharide (LPS)-induced activation. These data suggested that ECF administration by *i*.*n*. promotes activation of mLN NK cells.

### 2.4. ECF Induced IFN-γ and TNF-α Production in T Cells

Mature DCs induce differentiation and activation of naïve T cells [28,43]. Our data show that *i*.*n*. treatment with ECF induced DC activation in the mLN prompted the examination of effector T cell differentiation with ECF treatment. Intracellular producing levels of IFN-γ in mLN T cells were significantly increased with ECF treatment in the both C57BL/6 and BALB/c mice compared to that of the PBS-treated controls, although it was much lower than that of the LPS treatment (Figure 3A,B and Appendix A). Moreover, TNF-α producing levels in the mLN CD8 and CD4 T cells were also substantially increased with ECF in both C57BL/6 and BALB/c mice, which was similar to that in the LPS treatment (Figure 3A,B and Appendix A). Thus, these data suggested that *i*.*n*. administration of ECF promoted the production of IFN-γ- and TNF-α in CD4 and CD8 T cells in mLN.

### 2.5. Activation of DCs by ECF Elicited T and NK Cell Activation

As mentioned above, NK and T cell activation are controlled by mature DCs. Next, whether the ECF-stimulated DCs were essential for induction of T and NK cell activation was examined. DCs were depleted using biotin-conjugated anti-CD11c antibodies (Abs) in the mLN cell suspension (Figure 4A), and the cells were treated with ECF. The elevated levels of CD69 and TRAIL in NK cells by ECF were dramatically reduced in DC-depleted mLN cells (Figure 4B). Moreover, ECF-induced production of granzyme B, IFN-γ, and perforin was diminished in DC-depleted mLN cells (Figure 4C). Furthermore, the elevated levels of IFN-γ- and TNF-α-producing CD8 and CD4 T cells with ECF treatment were almost completely absent in DC-depleted mLN cells (Figure 4D). These data demonstrated that activation of NK and T cells by ECF was mediated by activated DCs in the mLN.

### 2.6. ECF Enhanced Anti-Cancer Effect of Anti-PD-L1 Antibody

The finding that ECF promoted immune activation in the mLN prompted the evaluation of the adjuvant effect of ECF in enhancing the anti-PD-L1 Abs effect for cancer immunotherapy. The C57BL/6 mice were inoculated intravenously (*i*.*v*.) with B16 melanoma cells to establish metastatic lung cancer. Treatment with ECF *i*.*n*. slightly delayed death of the mice compared to that of the PBS controls (Figure 5A). The anti-PD-L1 treated mice died within 23 days after tumor challenge, which was delayed the cancer-related death of the mice compared to that of the PBS-treated mice (Figure 5A). The combination of ECF and anti-PD-L1 Abs treated mice survived up to 30 days of tumor injection (Figure 5A). The body weight of mice that were treated with the combination of ECF and anti-PD-L1 Abs slightly increased, while the control-treated mice showed a gradual decrease in body weight (Figure 5B). Moreover, ECF plus anti-PD-L1 Abs treatment inhibited B16 cell infiltration into the lung, whereas other treatments did not (Figure 5C). These results suggested that ECF promoted NK and T cell activation; therefore, the contribution of NK and T cells to the ECF-induced anti-tumor effect was examined. As shown in Figure 5D, NK or T cell depleted mice failed to prevent the death of mice after *i*.*v*. injection of B16 tumor cells. On the day of survival, the mice were much shorter in the group depleted of both NK and T cells compared to other controls (Figure 5D). Next, we evaluated the effect of ECF in the different type of tumor, CT-26 carcinoma, in BALB/c mice. The metastatic CT-26 tumor growth in the lung was almost completely prevented by the combination of ECF and anti-PD-L1 antibody, while the anti-PD-L1 Abs or ECF alone treated mice were shown CT-26 cell infiltration and tumor growth in the lung (Figure 5E,F). Therefore, these data demonstrated that ECF enhanced the anti-tumor effect of anti-PD-L1 antibody by activating NK cells and CTLs.

## 3. Discussion

Fucoidans, the marine sulfated polysaccharides, have complicated and heterogenous structures due to the source and extraction procedure [4]. The factors, such as sugar monomers, molecular masses, sulfate ester pattern and content are involved within their potential activities [44]. Different composition of monosaccharide in the polysaccharide may elicit different biological activity. Study showed that the sulfation pattern was one of the important factors for the biological activity of fucoidan [45,46]. In this study, we detected the presence of two sulfate groups in ECF, which might show the functional structure. We found that *i*.*n*. administration of ECF induced immune cell activation including DCs, NK cells, and T cells in the mLN. Additionally, the study showed that NK and T cell stimulation by ECF was mediated by activation of DCs, as shown by the deletion of DCs in the mLN, which failed to promote protective immune responses. The ECF plus anti-PD-L1 Ab treatment prevented tumor cell growth in the lung by activating CTLs and NK cells. Therefore, this study demonstrated that ECF functions as a mucosal immune stimulator to enhance the anti-cancer activity of anti-PD-L1 Abs against B16 and CT-26 tumors.

Immune-modulating properties have been reported with marine natural polysaccharides, including ascophyllan, carrageenan, fucoidan, and porphyran, which have effectively stimulated an immune response against cancer, as well as treatment of other infectious diseases [8,17,47,48]. Seaweed fucoidans have been explored for their mechanistic activities of immunomodulation; *F*. *vesiculosus*-extracted fucoidan induces DC activation in mice and humans, thereby enhancing T cell immunity, which further induces anti-tumor immunity [6,17]. Fucoidans contain various types of monosaccharides such as fucose, xylose, mannose, glucose, and galactose [17,20]. In our previous study, the ECF adjuvant effect in combination with a carcinoma self-antigen has been explored, which protects mice from developing B16 melanoma [41]. The ECF contained higher content of galactose and fucose compared to other fucoidans [20]. Moreover, the ECF showed the most powerful effect in the stimulation of human DCs and T cells than other fucoidans [20]. However, its mucosal immunity is unknown; therefore, in this study, the hypothesis that ECF may be used as a mucosal adjuvant and further exhibits mucosal immunity has been investigated [7,41].

Activated CTLs and NK cells are involved in immunotherapy against cancer [27,49]. Activated, these cells’ pro-inflammatory cytokines and cytotoxic mediators accelerate the killing of target cells. In this study, we showed that intranasal treatment with ECF upregulated NK cell activation markers in the mLN, i.e., CD69 and TRAIL, and improved the production of pro-inflammatory cytokines and cytotoxic mediators in the BAL fluid compared to that of the PBS-treated group. Thus, this observation indicated that *i*.*n*. treatment of ECF elicited the NK cell activation in the mLN. Furthermore, in another set of experiments, ECF promoted IFN-γ production in CD4 and CD8 T cells in the mLN. Importantly, the ECF plus anti-PD-L1 antibody induced suppression of metastatic lung cancer growth was completely diminished by depletion of CD8 and NK cells. Thus, ECF prompted the CTL- and NK cell-mediated anti-cancer immune responses and acted as an immunostimulatory molecule.

Pattern recognition receptors (PRRs) are mainly expressed by APCs to contribute to their activation [50]. PRRs have been targeted by immune stimulatory molecules, such as scavenger receptors (SRs) and toll-like receptors (TLRs) [51,52,53]. Natural polysaccharides stimulate PRRs, such as polysaccharides from *Rehmannia glutinosa*, which is a natural herb that activates DCs in humans and mice via TLR4 stimulation [54]. Fucoidan, isolated from another species of brown seaweed, *F*. *vesiculosus*, induces human DC activation via SR-A [52]. Different species of extracted fucoidan have different immune modulatory and biological activities, possibly due to their different compositions [19]. Therefore, further studies are needed to compare and analyze the immunological stimulatory effects according to the molecular weight and monosaccharide composition of fucoidans between different species.

Immune checkpoint blockade therapies have shown promising outcomes in clinical trials in various types of cancers [34,38,39]. The PD-1 is expressed on T cells which binds to PD-L1 on tumor cells as well as APCs [36]. It has well investigated that anti-PD-1 Ab can stimulate T cells for induction of anti-cancer immunity [55]. In case of anti-PD-L1 Ab, DCs are the main inducer for activation of T cells [56]. It has been shown that the DCs are essential for PD-L1 blockade and the DCs upregulate expression of PD-L1 upon antigen uptake [56]. Since the anti-PD-L1 Ab targets DCs for induction of anti-cancer immunity, we combined the ECF with anti-PD-L1 Ab to enhance the anti-cancer immunity. Therefore, this study demonstrated that the upregulation of costimulatory and MHC molecules together with blockade of PD-L1 on DCs can improve induction of T cell mediated anti-cancer immunity. In the current study, mucosal immune responses induced by *i*.*n*. ECF administration were explored. The demand for mucosal adjuvants has grown owing to the formulation of mucosal vaccines that enhance the immune response against targeted diseases. The delivery of vaccines through the *i*.*n*. route is a better method compared to the conventional method of injection, which causes traumatic pain in patients. Additionally, the nasal route achieves a better systemic bioavailability and is most accepted in terms of delivering a drug combined with a mucosal adjuvant [57,58]. The advantage of nasal immunization is that it elicits secretion of immunoglobulin A in the mucosal [59]. Nasal delivery of vaccines blocks pathogen entry, induces local microbial-specific immune responses, and may be a suitable immunization method because of the abundance of T, B, and plasma cells, which prompts the mucosal adaptive immune response [60]. Since the ECF elicits DC, T and NK cell activation by *i*.*n*. administration, it may be able to develop as a mucosal adjuvant for vaccines against respiratory viral and bacterial infection. However, although the ECF can induce mucosal immune activation, it has the potential to induce pulmonary inflammation and autoimmune diseases as well. Therefore, further studies on side effects such as induction of autoimmune diseases by the ECF will be needed.

Although the immune checkpoint blockade showed a promising effect in cancer treatment, combination therapies are focusing on overcoming resistance and low efficacy of the immune checkpoint blockade [61]. Among various types of cancer therapeutic methods, immune stimulatory adjuvant treatment has received great attention, since the stimulator can enhance anti-cancer activity of the immune checkpoint inhibitors [61,62].

## 4. Materials and Methods

### 4.1. Ethics Statement

C57BL/6 (5–8 weeks old, female) and BALB/c (5–8 weeks old, female) mice were acquired from Shanghai Public Health Clinical Center (SPHCC, Shanghai, China) or purchased from Hyochang Science (Daegu, Korea). Mice were housed at the Laboratory Animal Center of SPHCC or Yeungnam University under specific-pathogen-free conditions. The studies were approved by the Yeungnam University Laboratory Animal Center (#2019-029, approved: 2019-11-29) and SPHCC (#2018-A049-01, approved: 2018). All experiments were performed based on basic animal ethical principles and were conducted according to the IACUC regulations of SPHCC and Yeungnam University.

### 4.2. Cancer Cell Line

The B16-F10 cells (ATCC, Manassas, VA, USA) were cultured in RPMI-1640 (Merck KGaA, Darmstadt, Germany) containing 1% of penicillin-streptomycin (Thermo Fisher Scientific, Waltham, MA, USA) and 10% of FBS (Sigma-Aldrich, St. Louis, MO, USA) under 5% CO_2_ at 37 °C. CT26.WT-iRFP-Neo (CT-26-iRFP; Imanis Life Sciences, CL091, Rochester, NY, USA) cells were cultured in 0.4 mg/mL of G418 (Sigma-Aldrich) contained DMEM medium.

### 4.3. Preparation of ECF

As described previously, ECF was purified and isolated [18,20,41]. Briefly, an organic solvent was added to 20 g of dried *E*. *cava* for overnight soaking. After drying, the cleaned ECF was mixed with 200 mL of 85% ethanol and stirred at 70 °C for 2 h. The samples were centrifuged (10 °C, 7000× *g* for 10 min) and then washed with acetone. The dried sample was incubated with HCl (0.01 N) at 80 °C for 2 h. The sample was then centrifuged at 8000× *g* at 25 °C for 10 min. The collected supernatant was mixed with CaCl_2_ (1%) at 4 °C for 18 h. The mixture was centrifuged at 8000× *g* for 10 min, and then the supernatant was harvested and mixed with ethanol in a 2:3 ratio. The mixture was centrifuged at 10,000× *g* for 30 min, dialyzed against distilled water (MWCO, 10–12 kDa), and lyophilized. The samples were dried, and the powder was used as the ECF.

### 4.4. Characterization of ECF

The weight-average molecular weight and polydispersity index of ECF were analyzed using HPLC (Agilent Technologies, Santa Clara, CA, USA), combined to GPC. The monosaccharide composition of ECF was carried out on a HPLC. FT-IR spectrum of ECF was analyzed between 500 and 3500 cm^−1^ by FT-IR spectrophotometer (Perkin Elmer Inc., Wellesley, MA, USA) at Core Research Support Center for Natural Products and Medical Materials in Yeungnam University.

### 4.5. Reagents and Antibodies

Isotype control antibodies and blockade of Fc receptor Abs were provided from BioLegend (San Diego, CA, USA). Anti-CD3 (17A2), anti-CD11c (N418), anti-CD40 (HM40-3), anti-CD49b (DX5), anti-CD80 (16-10A1), anti-CD69 (H1.2F3), anti-CD86 (GL-1), anti-CD253 (TRAIL, N2B2), anti-granzyme B (GB11), anti-IFN-γ (R4-6A2), anti- IL-6 (MP5-20F3), anti-IL-12/23p40 (C17.8), anti- MHC class I (AF6-88.5), anti-MHC class II (M5/114.15.2), anti-NK1.1 (PK136), anti-perforin (S16009A), anti-TCR-β (H57-597) and anti-TNF-α (MP6-XT22) were purchased from BioLegend (San Diego, CA, USA). Anti- PD-L1 Abs were purchased from BioXcell (Lebanon, NH, USA). LPS (O111:B4) was purchased from Sigma-Aldrich.

### 4.6. Preparation of a Single Cell Suspension of the Mediastinal Lymph Node (mLN)

As described previously [7,30,41,63], mLNs were harvested and grinded by slid glass. Through a 100 nm nylon mesh, the aggregated cells were removed and the mLN cell were digested with digestion buffer which contained with DNase I (Sigma-Aldrich) and collagenase type IV (Sigma-Aldrich). After washing with PBS, the cells were re-suspended with 5 mL of histopaque-1077 (Sigma-Aldrich) and upper layered with 5 mL of fresh histopaque-1077, and followed density cut by centrifugation at 2000× *g* for 10 min. The cells heavier than 1.077 were harvested as the leukocytes and suspended PBS.

### 4.7. Analysis of mLN DC Activation

The mLN cells were incubated with Fc blocking and control IgG Abs for 15 min and followed staining with surfaced mAbs; lineage Abs (anti-B220, anti-CD3, anti-CD49b, anti-CD90.1, anti-Gr-1, and anti-TER-119), costimulatory (anti-CD40, anti-CD80, and anti-CD86) and MHC class molecules (MHC class I and MHC class II) and anti-CD11c Abs. The DCs in mLN was determined as CD11c^+^lineage^−^ cells and the expression levels of CD40, CD80, CD86, MHC class I and II were analyzed in CD11c^+^lineage^−^ cells.

### 4.8. Activation of NK Cells

The mLN cells were incubated with Fc receptor-binding Abs, followed by staining with unconjugated isotype control Abs. For C57BL/6 mice, the mLN cells were incubated with CD3, CD69, NK1.1, and TRAIL Abs and the NK cells were defined as NK1.1^+^CD3^−^ cells. In case of BALB/c mice, the mLN cells were stained with CD49b and TCR-β Abs and CD49b^+^TCR-β^−^ cells were defined as the NK cells in BALB/c mice. After staining with 4,6-diamidino-2-phenylindole (DAPI; Sigma-Aldrich) solution, the cells were analyzed by flow cytometer (ACEA Biosciences Inc., San Diego, CA, USA).

### 4.9. Analysis of Intracellular Cytokine and Cytotoxic Mediator Production

The mLN single-cell suspension was incubated at 37 °C with 2 μM of monensin (BioLegend) for 2 h. The cells were stained dead cell detection reagents (Zombie Violet Fixable Viability Kit; BioLegend). The cells were stained with surface Abs at 4 °C for 15 min. After removing free Abs by centrifugation, the cells were fixed with fixation buffer (BioLegend) at 4 °C for 20 min and followed permeabilization by permablilization buffer (BioLegend). The cells then incubated with intracellular Abs at 25 °C for 20 min. After washing with PBS, cells were analyzed with a flow cytometer (ACEA Biosciences Inc.).

### 4.10. Enzyme-Linked Immunosorbent Assay (ELISA)

The levels of IFN-γ, IL-6, IL-12p70, and TNF-α were measured using ELISA kits (BioLegend). The ELISA kit for perforin was obtained from Abbkine (Wuhan, China). The ELISA kit for granzyme B was obatined from LSBIO (Seoul, Korea). The concentrations of these cytokines were quantified in triplicate.

### 4.11. Depletion of DCs

The single cell suspension of mLN cells were prepared and incubated with biotin-conjugated anti-CD11c Abs (BioLegend) at 4 °C for 15 min. After washing with PBS, the cells were incubated with anti-biotin-microbead (Miltenyi Biotec) for 20 min. CD11c-negative cells were harvested by magnetic selection method (Miltenyi Biotec). More than 95% of CD11c^+^ cells were depleted, as analyzed by flow cytometry (ACEA Biosciences).

### 4.12. Mouse Metastatic Lung Cancer Model and ECF Treatment

C57BL/6 mice were intravenously (*i*.*v*.) administered with B16 (0.5 × 10⁶) cells. BALB/c mice were *i*.*v*. transplanted with CT-26-iRFP (0.5 × 10⁶) cells. The tumor injected mice were randomly divided into PBS, anti-PD-L1 Abs, ECF, and anti-PD-L1 Abs + ECF groups (5 mice per group). We administrated ECF and anti-PD-L1 Ab to mice on different schedules and injection routes. ECF (50 mg/kg) was *i*.*n*. administered every 3 days from day 3 after tumor injection. For anti-PD-L1 Ab treatment, the mice received intraperitoneally (*i*.*p*.) with 10 mg/kg of anti-PD-L1 Abs every 3 days from day 5 after tumor injection. For survival rate, the mice were monitored till day 35 after B16 tumor injection (5 mice per group).

### 4.13. Histological Analysis

On day 10 after tumor challenge, 4% of paraformaldehyde (1 mL) infused to lungs and then harvested. The lungs were then fixed (4% of paraformaldehyde for 24 h) and followed paraffin embedding. The lung was sectioned to a thickness of 5 μm. After de-paraffinization and re-hydration of the sections, the sections were stained with hematoxylin and eosin (Sigma-Aldrich).

### 4.14. NK Cell and CD8 Cell Depletion

C57BL/6 mice were treated with 50 μg anti-NK1.1 (PK136) or anti-CD8α (2.43) Abs every two days (BioXcell), respectively. Two days after first injection of depletion Abs, the mice were injected *i*.*v*. with 0.5 × 10^6^ B16 cells. The depletion efficacy of NK or CD8 cells was higher than 95%.

### 4.15. In Vivo Fluorescence Imaging

Metastatic CT-26-iRFP cancer in lung was imaged using the Fluorescence In Vivo Imaging System, FOBI (Cellgentek, Cheongju, Korea) on day 14 after CT-26-iRFP challenge in BALB/c mice.

### 4.16. Statistical Analysis

Experiments were performed with two samples, each in triplicate. Data are analyzed as the mean ± standard error of the mean (SEM). Tukey multiple comparison test and the Mann-Whitney t-test were used to analyze all experimental data. *p* < 0.05 was considered significantly different.

## 5. Conclusions

In this study, we found that intranasal administration of ECF promoted activation of DC, T and NK cells in the mLN. Collectively, this study demonstrated that ECF functioned as a mucosal immune stimulator for enhancing the anti-cancer activity of immune checkpoint inhibitors against lung metastasis of melanoma and carcinoma.

## Figures and Tables

**Figure 1 ijms-22-09125-f001:**
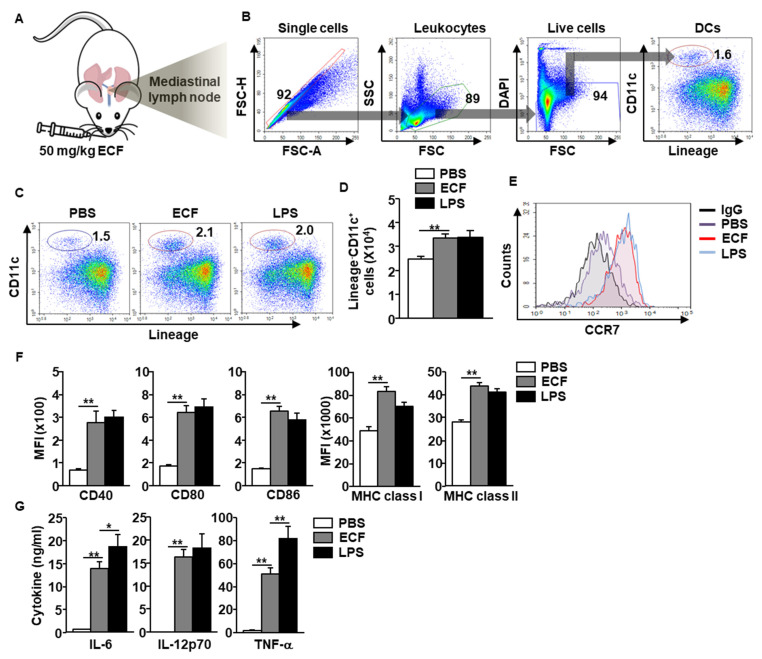
Dendritic cell (DC) activation in the mediastinal lymph node (mLN) by intranasal administration of *Ecklonia cava* fucoidan (ECF). (**A**) C57BL/6 mice were intranasally (*i*.*n*.) administered with 50 mg/kg ECF. The DC activation in mLN was harvested and analyzed 18 h after ECF administration. (**B**) The definition of DCs in mLNs was shown. (**C**) Percentage of mLN DCs, as analyzed by flow cytometry. PBS, phosphate-buffered saline; LPS, lipopolysaccharide. (**D**) Absolute number of DCs in the mLN. (*n* = 6, two-way analysis of variance, ** *p* < 0.01) (**E**) Expression levels of C-C chemokine receptor type 7 (CCR7) in mLN DCs were shown. IgG, immunoglobulin G. (**F**) Costimulatory molecules and major histocompatibility complex (MHC) class I and II expression levels were measured in mLN DCs. MFI, mean fluorescence intensity; (**G**) The levels of interleukin (IL)-6, IL-12p40, and tumor necrosis factor (TNF)-α levels in bronchoalveolar lavage (BAL) fluid. (*n* = 6, two-way analysis of variance, * *p* < 0.05 and ** *p* < 0.01).

**Figure 2 ijms-22-09125-f002:**
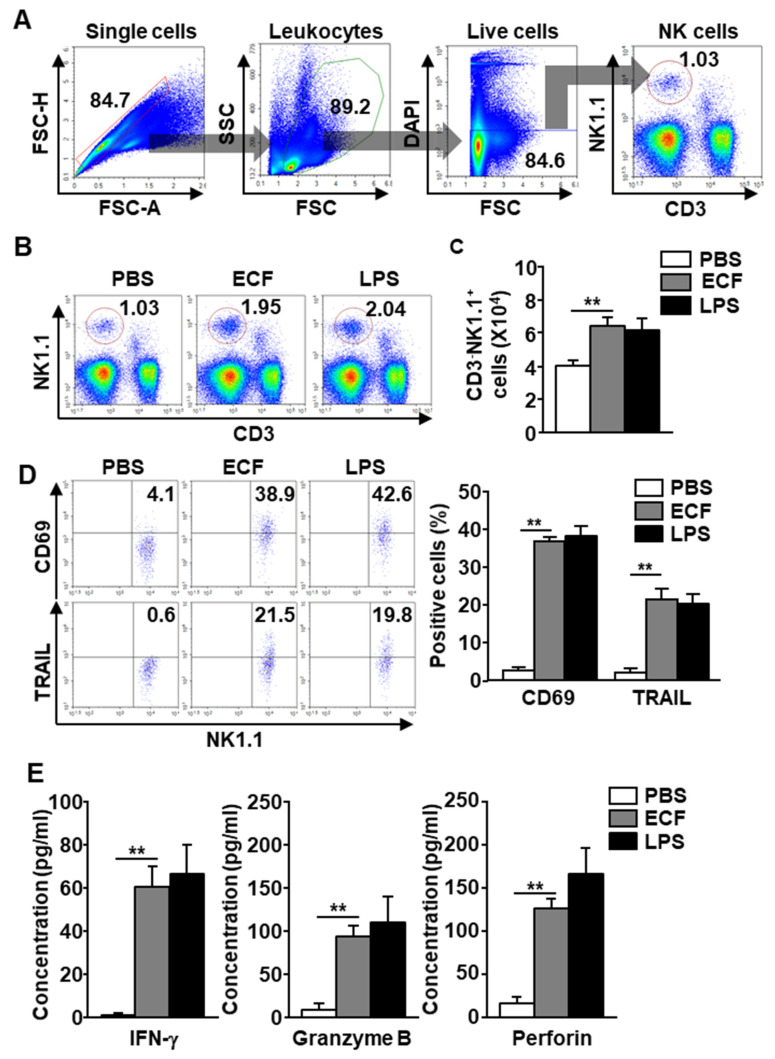
Natural killer (NK) cell activation by ECF. Twenty-four hours after 50 mg/kg ECF administration intranasally, the mLN was harvested from C57BL/6 mice. (**A**) NK cells definition was shown. (**B**) The percentage of NK cells was shown. PBS, phosphate-buffered saline; LPS, lipopolysaccharide. (**C**) Average of NK cell number in the mLN (*n* = 6, two-way analysis of variance, ** *p* < 0.01). (**D**) Surface activation markers, CD69 and tumor necrosis factor-related apoptosis-inducing ligand (TRAIL) expression levels in mLN NK cells were shown (**left panel**). Mean number of CD69 and TRAIL positive cells in mLN NK cells were shown (**right panel**, *n* = 6, two-way analysis of variance, ** *p* < 0.01). (**E**) Concentration of interferon (IFN)-γ, granzyme B, and perforin levels in BAL fluid was shown. (*n* = 6, two-way analysis of variance, ** *p* < 0.01).

**Figure 3 ijms-22-09125-f003:**
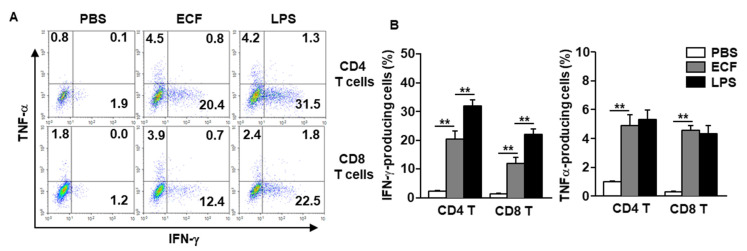
Induction of T helper 1 and cytotoxic T 1 (Tc1) immune responses in the mLN by intranasal administration with ECF. C57BL/6 mice were *i*.*n*. administered with 50 mg/kg ECF twice at a 3-day interval. (**A**) Intracellular producing levels of IFN-γ and TNF-α in mLN CD4 and CD8 T cells were measured. (**B**) Mean percentage of IFN-γ-producing (**left panel**) and TNF-α-producing (**right panel**) CD4 and CD8 T cells (*n* = 6, two-way analysis of variance, ** *p* < 0.01).

**Figure 4 ijms-22-09125-f004:**
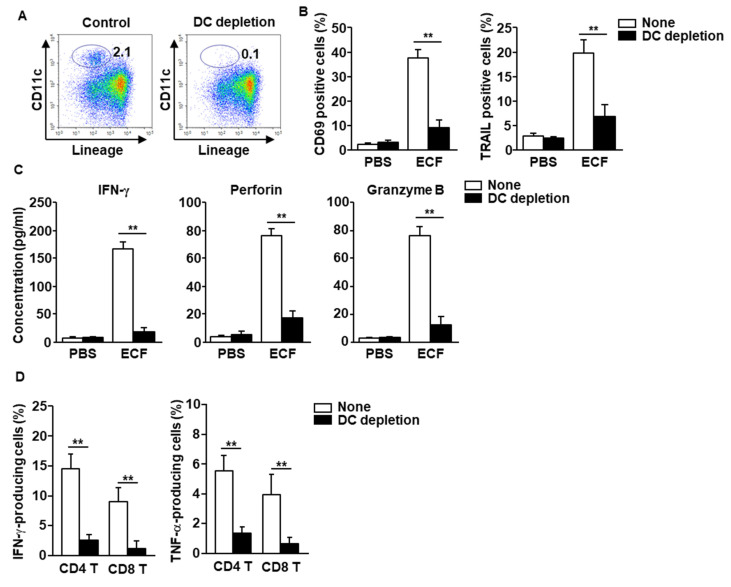
DCs are required for stimulation of NK and T cells in the mLN. (**A**) DCs in mLN cells were depleted, as described in the Methods. (**B**) CD69 and tumor necrosis factor-related apoptosis-inducing ligand (TRAIL) expression levels, as measured 24 h after treatment with 100 μg/mL of ECF. PBS, phosphate-buffered saline (*n* = 6, two-way analysis of variance, ** *p* < 0.01). (**C**) The concentration of IFN-γ, perforin, and granzyme B in the cultured medium was shown. (**D**) Percentage of intracellular IFN-γ- and TNF-α-producing T cells in mLN cells after culture with or without 100 μg/mL ECF for 3 days, as analyzed by flow cytometry (*n* = 6, two-way analysis of variance).

**Figure 5 ijms-22-09125-f005:**
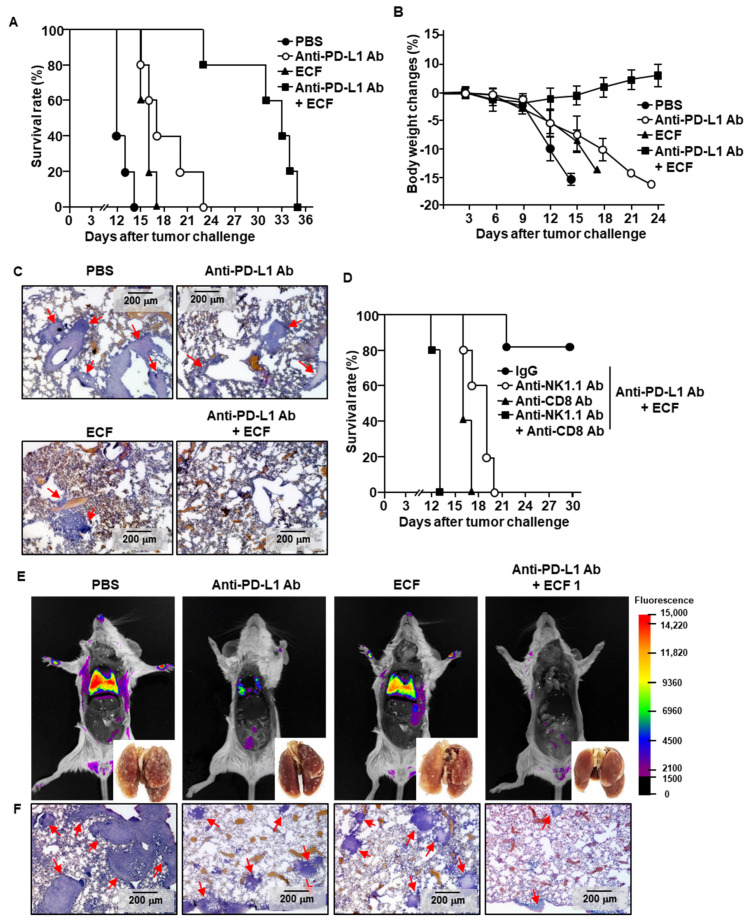
ECF enhances anti-PD-L1 antibody-induced anti-cancer immunity. C57BL/6 mice were challenged intravenously (*i*.*v*.) with 0.5 × 10^6^ B16 cells. As indicated in the materials and method, the mice were treated with PBS 10 mg/kg of anti-PD-L1 antibody (Ab), 50 mg/kg of ECF and the combination of ECF and anti-PD-L1 Ab. (**A**) The graph showed survival rate of treated mice (*n* = 5 for each group). (**B**) Changes of body weight during treatment were shown. (**C**) Hematoxylin and eosin (H&E) staining of lung showed B16 melanoma cell infiltration, as analyzed on day 10 after tumor injection. Red arrows indicated infiltrated tumor cells. (**D**) Survival rate, as measured in natural killer (NK)- or CD8 T cell-depleted mice (*n* = 5 for each group). (**E**,**F**) BALB/c mice were injected *i*.*v*. with 0.5 × 10⁶ CT-26-iRFP cells. The mice received with PBS, 10 mg/kg of anti-PD-L1 Ab, 50 mg/kg of ECF or the combination of anti-PD-L1 Ab and ECF for a 3 day-interval. (**E**) Fluorescence imaging of iRFP in the mice on day 14 after CT-26-iRFP cell administration in BLAB/c mice, *n* = 5. (**F**) CT-26 cell infiltration in lung was analyzed by H&E staining of lung tissue 14 days after CT-26 injection in BALB/c mice. Red arrows indicated infiltrated tumor cells.

**Table 1 ijms-22-09125-t001:** The component analysis of *Ecklonia cava* fucoidan (ECF). The molecular weight (M_w_) was determined by gel permeation chromatography (GPC). The monocyte composition of ECF has analyzed by HPLC. Turbidimeric assay was conducted for determination of SO_4_^2−^ in ECF. The numbers show the percentage of the component in the ECF.

	M_W_ (kDa)	PDI	Composition of Monosaccharide (%) ^a^					SO_4_^2−^^b^
			Fuc	Xyl	Glu	Man	Gal	
*Ecklonia cava* fucoidan	42.95	1.39	45.13	1.78	1.37	0.55	23.24	26.46

^a^ Determined using HPLC analysis after acidic hydrolysis; ^b^ Determined using turbidimetric assay after acidic hydrolysis. Abbreviation: polydispersity index, PDI; Fuc, fucose; Xyl, xylose; Glu, glucose; Man, mannose; Gal, galactose.

## Data Availability

The data are available in author’s archive.

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
