# Peer review of "Enhancement of Immune Checkpoint Inhibitor-Mediated Anti-Cancer Immunity by Intranasal Treatment of Ecklonia cava Fucoidan against Metastatic Lung Cancer"

_ijms, 2021, doi:10.3390/ijms22179125_

Round 1

Reviewer 1 Report

Authors use fucoidan as an immuno stimulatory molecule to use in combination with PD1 treatment to counter B16-F10 tumors and others.  The composition of the fucodan prep is not discussed in detail. There is some HPLC analysis but one would like to see a detailed analysis of what is given to the mice and or what part of that prep is the active reagent causing these observations.  The immune effects on the NK and DC cells is not as clear. There is not much difference between fucodan and LPS. The mouse tumor treatment experiments are interesting but none of the studies have a positive control to compare to.  Would LPS work as well? Why the authors used anti PDL1 antibody is not discussed, a PD1 blocking antibody would be more appropriate.  The tumor treatments were started after 3 days of tumor inoculation. Most patients would be getting the treatment after they developed  tumor. So a more detailed tumor treatment would be more informative. Mice should be treated immediately after tumor innoculation, 3 days and maybe after the first sight of palpable tumor. Animals should be kept longer to see if the survival is stable since in Fig 5E one can still see tumors on inflated lungs of the combination treated mice.

Author Response

Reviewer 1

Authors use fucoidan as an immuno stimulatory molecule to use in combination with PD1 treatment to counter B16-F10 tumors and others. 

Answer (A): Thank for the crucial comments.

The composition of the fucodan prep is not discussed in detail.

A: We thank the reviewer’s constructive comments. We have now discussed the composition of fucoidan in Discussion part.

There is some HPLC analysis but one would like to see a detailed analysis of what is given to the mice and or what part of that prep is the active reagent causing these observations. 

A: As the reviewer mentioned, we agreed that the biological activities of fucoidan may be highly correlated with the structural and nonstructural features. Since studies have showed the biological activities of fucoidan isolated from different marine seaweeds different effect as well as those fucoidans have various structure factors, including molecular size, monosaccharide and sulfate contents. In this study, we harvested polysaccharide peaks from HPLC analysis and evaluated immune stimulatory effect by intranasal administration. As shown the molecular weight and composition of ECF in Table 1, the fucoidan contained fucose, xylose, glucose, manose and galactose. Furthermore, we showed the detail presence of sulfate groups by FT-IR (Supplementary Figure 1). The ECF was composed higher amount of fucose and sulfated ester groups compared to other fucoidans. It has shown that higher amount of fucose and sulfated ester groups may be mainly contribute biological activities of the fucoidan such as anti-tumor and anti-inflammatory effects [Revewers References 1-4].

The immune effects on the NK and DC cells is not as clear.

A: Immune stimulatory effect of ECF has been shown in Figure 1 and 2. The DC population in the spleen has clearly defined by flow cytometry. As the reviewer knows that the spleen contained several types of immune cells, which were gated out by lineage staining. Among them, only CD11c-positive cells were analyzed as the DCs. Moreover, the NK cell population also clearly defined by CD3 and NK1.1. The method of DC analysis has already reported many different paper [Revewers References 5-9]. 

There is not much difference between fucodan and LPS.

A: LPS is the most well-known immune stimulatory molecules. Since the ECF also promoted immune activation, the effect in the immune activation by ECF can be similar with LPS. However, the LPS induced much increased production levels of inflammatory cytokine in the BAL fluid and CD4 T cells compared to ECF.

The mouse tumor treatment experiments are interesting but none of the studies has a positive control to compare to. Would LPS work as well?

A: Unfortunately, there is no positive control to compare with ECF. We though the LPS can be positive control, however continuous treatment of LPS promoted death of the mice by inflammation. Therefore, we could not get data from LPS-treated mice as positive control.

Why the authors used anti PDL1 antibody is not discussed, a PD1 blocking antibody would be more appropriate. 

A: We have revised the discussion section. As the reviewer knows that anti-PD-1 and anti-PD-L1 Ab has different mechanism for induction of anti-cancer immunity. The anti-PD-1 directly stimulate T cells since the T cells mainly expresses the PD-1 on their surface. However, the PD-L1 is detected tumor cells as well as antigen presenting cells. Many people though that anti-PD-L1 interrupt interaction of tumor cell and T cell, which might elicit anti-cancer immunity by prevention of tolerance. However, recent study has demonstrated that anti-PD-L1 Ab-induced anti-cancer effect was mediated by DCs. Since the anti-PD-L1 Ab blocked interaction with T cells, which induces activation of the T cells [32973173]. Meanwhile, the remained circulating anti-PD-L1 attached on tumor cells, eventually the T cell can kill the tumor cells. Therefore, the combination of ECF and anti-PD-L1 Ab can be targeted activation of DCs, which increased anti-cancer immunity.

The tumor treatments were started after 3 days of tumor inoculation. Most patients would be getting the treatment after they developed tumor. So a more detailed tumor treatment would be more informative. Mice should be treated immediately after tumor innoculation, 3 days and maybe after the first sight of palpable tumor.

A: As the reviewer mentioned, the patient will receive treatment of the anticancer drugs after developed tumor. In general, we treated mice with reagents 7 days after subcutaneous injection of tumor cell when the tumor could be detected under skin. Since lung metastatic tumor developed faster than subcutaneously injected tumor, we treated mice 3 days after tumor injection. The 3 days may be the first sight of palpable tumor as the reviewer mentioned, however it cannot clearly demonstrated since it was lung metastatic tumor. We have revised the materials and method sections.

Animals should be kept longer to see if the survival is stable since in Fig 5E one can still see tumors on inflated lungs of the combination treated mice.

A: We appropriate the critical comments. Yes, the mice treated with combination of ECF and anti-PD-L1 Ab were also developed tumor in the lung and the mice has died within 35 days after tumor injection. We have now revised the Figure 5A.

Reviewers references.

  1. Apostolova, E.; Lukova, P.; Baldzhieva, A.; Katsarov, P.; Nikolova, M.; Iliev, I.; Peychev, L.; Trica, B.; Oancea, F.; Delattre, C.; Kokova, V., Immunomodulatory and Anti-Inflammatory Effects of Fucoidan: A Review. Polymers (Basel) 2020, 12, (10).
  2. Wang, Y.; Xing, M.; Cao, Q.; Ji, A.; Liang, H.; Song, S., Biological Activities of Fucoidan and the Factors Mediating Its Therapeutic Effects: A Review of Recent Studies. Mar Drugs 2019, 17, (3).
  3. Zhang, W.; Oda, T.; Yu, Q.; Jin, J. O., Fucoidan from Macrocystis pyrifera has powerful immune-modulatory effects compared to three other fucoidans. Mar Drugs 2015, 13, (3), 1084-104.
  4. Zhang, W.; Park, H. B.; Yadav, D.; Hwang, J.; An, E. K.; Eom, H. Y.; Kim, S. J.; Kwak, M.; Lee, P. C.; Jin, J. O., Comparison of human peripheral blood dendritic cell activation by four fucoidans. Int J Biol Macromol 2021, 174, 477-484.
  5. Zhang, W.; An, E.-K.; Park, H.-B.; Hwang, J.; Dhananjay, Y.; Kim, S.-J.; Eom, H.-Y.; Oda, T.; Kwak, M.; Lee, P. C.-W., Ecklonia cava fucoidan has potential to stimulate natural killer cells in vivo. International Journal of Biological Macromolecules 2021.
  6. Hwang, J.; Zhang, W.; Park, H.-B.; Yadav, D.; Jeon, Y. H.; Jin, J.-O., Escherichia coli adhesin protein-conjugated thermal responsive hybrid nanoparticles for photothermal and immunotherapy against cancer and its metastasis. J Immunother Cancer 2021, 9, (7).
  7. Zhang, W.; Xu, L.; Park, H. B.; Hwang, J.; Kwak, M.; Lee, P. C. W.; Liang, G.; Zhang, X.; Xu, J.; Jin, J. O., Escherichia coli adhesion portion FimH functions as an adjuvant for cancer immunotherapy. Nat Commun 2020, 11, (1), 1187.
  8. Park, H.-B.; Lim, S.-M.; Hwang, J.; Zhang, W.; You, S.; Jin, J.-O., Cancer immunotherapy using a polysaccharide from Codium fragile in a murine model. OncoImmunology 2020, 9, (1), 1772663.
  9. Park, H.-B.; Hwang, J.; Lim, S.-M.; Zhang, W.; Jin, J.-O., Dendritic cell-mediated cancer immunotherapy with Ecklonia cava fucoidan. International Journal of Biological Macromolecules 2020, 159, 941-947.

Reviewer 2 Report

The manuscript “Enhancement of Immune Checkpoint Inhibitor-Mediated Anti-Cancer Immunity by Intranasal Treatment of Ecklonia cava Fucoidan against Metastatic Lung Cancer” by Zhang et al. demonstrates an interesting finding of immune activation in lung cancer. The authors further showed that the activity of ECF as a natural immune adjuvant for PD-L1 immunotherapy. Even though the work is interesting, there are some minor concerns in the study which need to be addressed before consideration for publication. Those are listed below.

  1. Did the authors check for any adverse effects from ECF and anti-PD-LI therapy apart from body weight analysis?
  2. The authors propose ECF as an immune stimulant. However, did mice show any signs of high levels of Immunoglobulins? Especially IgA? Did they check for any signs of autoimmune disease? If not, at least they need to discuss this in the discussion.
  3. The authors need to provide high-resolution images for Figures 5C and F. They also need to give better clarifications for these figures. Eg: how to identify tumor cell infiltration on H&E stained slides
  4. The authors need to correct the inappropriate use of some words, titles and abbreviations in the manuscript.

Eg:

Page 2 line 81 - …he hypothesized needs to be corrected

Page 3 line 99 – Table 1 needs a proper caption

Page 3 line 105 – i.n.

Page 10 line 299 and 301 – check point should be checkpoint

Author Response

Reviewer 2

The manuscript “Enhancement of Immune Checkpoint Inhibitor-Mediated Anti-Cancer Immunity by Intranasal Treatment of Ecklonia cava Fucoidan against Metastatic Lung Cancer” by Zhang et al. demonstrates an interesting finding of immune activation in lung cancer. The authors further showed that the activity of ECF as a natural immune adjuvant for PD-L1 immunotherapy. Even though the work is interesting, there are some minor concerns in the study which need to be addressed before consideration for publication. Those are listed below.

 Answer (A): We appropriate the reviewer’s valuable comments.

  1. Did the authors check for any adverse effects from ECF and anti-PD-LI therapy apart from body weight analysis?

A: We observed colon inflammation and jagged hair in the mice during treatment, but no specific changes were observed.

  1. The authors propose ECF as an immune stimulant. However, did mice show any signs of high levels of Immunoglobulins? Especially IgA? Did they check for any signs of autoimmune disease? If not, at least they need to discuss this in the discussion.

A: Since this study has focused for treatment of cancer, we did not measure immunoglobulin levels. However, we have now preparing one more research paper that the ECF can functions as mucosal adjuvant for eliciting covid19 vaccine activity. The intranasal administration of ECF combined with spike protein induces dramatic increases in the IgA levels in BAL fluid and IgG levels in the blood serum (Reviewers Figure 1). We are really sorry about that we could not include this data in this paper, therefore we added the data as reviewer figure.

As mentioned above, the mice did not show any signs of autoimmune disease and inflammation in the lung.

Reviewers Figure 1. Intranasal administration of ECF functions as an adjuvant for inducing spike antigen-specific humoral immunity. BALB/c mice were injected i.n. with ECF and spike protein on day 0, 15, 30. On day 35 after first injection of ECF and spike protein, the mice were sacrificed and BAL fluid and sera were harvested. Spike proteins specific IgA and IgG levels were analyzed by ELISA.

  1. The authors need to provide high-resolution images for Figures 5C and F. They also need to give better clarifications for these figures. Eg: how to identify tumor cell infiltration on H&E stained slides

A: We have now revised the data. The tumor infiltration in the lung has indicated by red arrows.

  1. The authors need to correct the inappropriate use of some words, titles and abbreviations in the manuscript.

A: We are sorry for the error and thanks for the pointing out the errors. We have now corrected the errors. 

Eg:

Page 2 line 81 - …he hypothesized needs to be corrected

A: We corrected this error.

Page 3 line 99 – Table 1 needs a proper caption

A: We have added the caption for Table 1.

Page 3 line 105 – i.n.

A: We corrected this error.

Page 10 line 299 and 301 – check point should be checkpoint

A: We corrected this error.
